# Understanding the needs of family caregivers of stroke patients with disabilities: A phenomenological study using the timing it right theory

Miaozhen Wang[1], Ke Wang[1], Bin Xie[1], Dan Liu[2], Lingling Song[1], Xing Cao[1], Yunhui Tong[1], Chenting Liu[1], Hongyu Fu[1], You Zhou[1], Qinqin Chen[3], Yan Zhu[4], Ling Zhu[4], Fangqun Cheng(iD)[1]*

1 Department of Nursing, Xiangtan Central Hospital, Xiangtan, China, 2 Xiangtan Medicine & Health Vocational College, Xiangtan, China, 3 Publicity Department, Xiangtan Central Hospital, Xiangtan, China, 4 Department of Neurology, Xiangtan Central Hospital, Xiangtan, China

* 445935584@qq.com

## Abstract

### Background

Stroke is a leading cause of disability worldwide, significantly impacting patients and their families. Family caregivers, as the primary providers of care for stroke patients, face dynamic and evolving challenges throughout the patient's recovery journey.

### Objective

This study aims to explore the comprehensive care needs of family caregivers of stroke patients with disabilities within the Chinese cultural context. The study is guided by the "Timing It Right" (TIR) theory, a conceptual model emphasizing dynamic support aligned with caregivers' evolving needs across distinct care phases.

### Methods

Using a phenomenological qualitative design, this study recruited 48 family caregivers of stroke patients with disabilities from a tertiary hospital in Xiangtan City, China, from February 2024 and November 2024. Data were collected through semi-structured interviews and were analyzed thematically using NVivo 12 software. This study explored caregivers' experiences and needs across five key phases of care: diagnosis, stabilization, discharge preparation, adjustment, and adaptation.

### Results

In this study, family caregivers of stroke patients with disabilities expressed diverse needs across different phases of the patient's illness. (1) Diagnosis phase: hospital environment adaptation, examination navigation and disease knowledge acquisition.

**Data availability statement:** The data supporting the findings of this study are not publicly available due to ethical considerations. The Ethics Committee of Xiangtan Central Hospital has imposed restrictions on sharing a de-identified dataset, as the qualitative data contain potentially identifying and sensitive information that could compromise participant privacy. In accordance with these ethical guidelines, data cannot be made openly accessible. Researchers seeking access to the minimal data set necessary to verify the findings of this study are advised to contact the Ethics Committee of Xiangtan Central Hospital to request permission via telephone (+86073158214988) or email (xtszxyy_llwyh@sina.com). Data requests will be reviewed by the Ethics Committee of Xiangtan Central Hospital in accordance with their established protocols to ensure participant confidentiality is maintained. Please include the manuscript title and authors in your request.

**Funding:** This work was funded by the Hunan Provincial Health Commission Project (202203073335 ), Department of Science and Technology of Hunan Province (2024ZK4233), Hunan Provincial People's Hospital Medical Union Special Project (2023YLT002), Natural Science Foundation of Hunan Province of China (2024JJ9561), Chinese Nursing Association Project (ZHKY202406), and National Key Clinical Specialties Major Specialty Program of the Healthcare Commission of Hunan Province (Z2023138),Xiangtan Medical Association Project (2023-xtyx-40).

**Competing interests:** The authors declare that they have no conflicts of interest.

(2) Stabilization phase: rehabilitation skill development and caregiver well-being support. (3) Discharge preparation phase: home care readiness and health behavior guidance. (4) Adjustment phase: sequelae management and care continuity needs. (5) Adaptation phase: obtaining follow-up resources and safety management. Caregivers' needs were influenced by cultural expectations(e.g., Chinese traditional filial piety norms requiring family-based caregiving), resource limitations, and their evolving role over time.

## Conclusion

The findings emphasize the necessity of developing culturally sensitive interventions that are specifically tailored to meet the needs of family caregivers across distinct phases of care. Consideration should be given to integrating multidisciplinary teams, leveraging telemedicine for continuity of care, and designing caregiver education programs aligned with Confucian family values. These strategies can reduce caregiver burden and improve stroke rehabilitation outcomes in resource-limited settings.

## 1. Introduction

Stroke is characterized by high incidence, disability rate, mortality, recurrence and significant economic burden, making it become the third leading cause of disability among adults worldwide. The absolute number of incident strokes globally increases by 70.0% over the three-decade period [1]. Approximately 70–80% of stroke survivors are left with varying degrees of disability, with one-third relying on others for assistance with daily activities [1,2], imposing significant caregiving burdens on families and society. The primary caregivers of stroke patients with disabilities are predominantly family members [3], who disproportionately bear the caregiving burdens. The chronic burdens of caregiving often lead to chronic physical tension in caregivers, manifesting as fatigue, sleep disorders, and pain, and depressive symptoms are reported by approximately 40% of these individuals [4]. Critically, caregivers of stroke survivors are often lack professional caregiving knowledge and skills to adequately address the complex and evolving long-term care needs of survivors, resulting in inconsistent care quality and negatively impacting the functional recovery of stroke survivors. To bridge this gap, they crave relevant guidance to meet care needs.

Caregivers' needs refer to their ability to benefit from healthcare services [5]. More often than not, they need targeted information on daily care tasks, psychological and financial support systems, and foundational training in post-stroke rehabilitation strategies. Studies indicate that failure to promptly address caregivers' needs exacerbates their caregiving burden, thereby compromising their caregiving capacity [4]. And poor caregiver care can actually reduce activities of daily living (ADLs) of stroke survivors by an average of 12% [6]. Understanding and meeting these needs can not only enhance caregivers' abilities but also improve the overall quality of life for families and indirectly promote patient recovery. Additionally, the recovery process

for stroke patients with disabilities can last months or even years [1], during which caregivers' needs undergo significant changes. Caregivers desire more theoretical knowledge and improved caregiving skills in the early phases of the patient's illness, and require more continuity of care and social support at home after discharge [7]. The content and priorities of caregiver needs differ significantly across stroke phases. The Timing It Right (TIR) theory [8] offers a practical framework for systematically addressing these dynamic and phase-dependent challenges. By dividing the care continuum into five phases (diagnosis, stabilization, discharge preparation, adjustment, and adaptation), it provides a structured approach to map caregivers' evolving priorities. Therefore, understanding caregivers' needs throughout different phases of the disease provides a scientific basis for developing targeted support policies and optimizing nursing interventions that directly address their unmet informational, psychological, and competency gaps to sustain caregiving continuity.

The Timing It Right Theory (TIR), proposed by Canadian scholar Cameron [8] and colleagues, has been demonstrated to effectively assess the needs of family caregivers at different phases of a patient's illness trajectory [9], aligning closely with the dynamic changes in caregiving demands during the recovery process of stroke patients. This theory is based on the different needs of caregivers' knowledge and rehabilitation skills at different phases of the disease. By identifying the specific needs of caregivers at different phases, TIR theory has been successfully applied to family caregivers of stroke patients [89], establishing a scientific foundation for delivering timely and targeted intervention strategies [8].

Previous studies [4,10,11] have primarily focused on the psychological well-being, financial burden, social support, and coping strategies of stroke caregivers. However, they predominantly adopt a static or fragmented perspective—either focusing on isolated caregiving phases or addressing needs as uniform across the recovery continuum. There is a lack of research on the dynamic changes in the needs of caregivers for stroke patients with disabilities throughout the entire disease trajectory. Caregivers may prioritize medical knowledge acquisition during early phases but face escalating challenges in safety management or emotional resilience as caregiving prolongs. Therefore, it is necessary to pay attention to the dynamic changes in the needs of caregivers. Few studies have investigated the interplay between Chinese cultural norms of filial piety, which emphasize family-led care, and systemic barriers, such as fragmented community rehabilitation services, in shaping the dynamic and evolving unmet needs of caregivers across the stroke care continuum, from acute hospitalization to long-term home adaptation. These dual gaps neglect both temporal evolution and cultural mediation of needs, hindering the development of tailored, context-sensitive interventions for diverse populations.

Therefore, based on the framework of the TIR theory, this study aims to explore the whole-course caregiving experiences of family caregivers of stroke patients with disabilities, identify the specific needs and challenges of caregivers in the changing dynamics of stroke, and provide a basis for the development of future intervention programs.

## 2. Methods

### 2.1. Design

This study used a phenomenological qualitative design to understand the experience of the family caregivers of stroke patients with disabilities and explore their care needs at different phases.

### 2.2. Participants and sampling

Family caregivers of stroke incapacitated patients who were hospitalized and discharged from the Department of Neurology of a tertiary hospital in Xiangtan City from February 2024 to November 2024 were selected. Potential participants were identified by the research team through daily screenings of patient admission records and consultations with attending neurologists. Eligible patients were then approached by a member of the research team, who provided them with a detailed explanation of the study objectives, procedures, and potential risks and benefits. Purpose sampling and maximum variation sampling were used. Recruitment continued until thematic saturation was achieved, defined as the point where no novel themes emerged from consecutive interviews. Each case corresponded to only one caregiver, and both patients and caregivers were required to meet the inclusion and exclusion criteria. The sample included caregivers from

both rural and urban settings, with diverse educational backgrounds and caregiving roles. This diversity enhanced the potential applicability of findings to caregivers in similar socioeconomic and cultural contexts within China.

Inclusion criteria of patient: (1) patients with first-ever stroke confirmed by cranial CT or MRI in accordance with updated American Heart Association and American Stroke Association criteria [12]; (2) age ≥ 18 years; (3) daily living scale score (Barthel's index)≤60 during hospitalization, and stabilized vital signs; and (4) home or community-based rehabilitation after discharge from the hospital. Patient exclusion criteria: (1) patients with severe cardiac, pulmonary, hepatic, and renal diseases; (2) patients whose conditions worsened during treatment and who could not continue treatment.

Inclusion criteria for family caregivers: (1) family members of the patient, including spouse, parents, children, daughter-in-law (son-in-law), siblings, etc.; (2) taking primary care of the patient during and after hospitalization (≥4 h per day; if there are more than one caregiver, the one who takes care of the patient for the longest phase of time will be selected); (3) aged ≥18 years old; (4) conscious, with good reading comprehension and communication and expression skills; and (6) voluntary participation. Exclusion criteria for family caregivers: (1) people who have participated in training related to stroke patient care; (2) paid caregivers such as domestic helper.

A total of 48 cases were interviewed across five phase. Since the start of this study, no shedding phenomenon has occurred. Interviews were conducted at the diagnosis phase (diagnosis was clear and started in general 3–7 d), the stabilization phase (the condition was stable, there was no new cerebral hemorrhage or cerebral infarction, and there was no obvious enlargement of the foci), the discharge preparation phase (from the time when the patient was about to finish the main treatment in the hospital to the time when he/she was discharged), the adjustment phase (3 months after discharge and going home), and the adaptation phase (3–6 months after discharge and going home). Each phase was denoted by the letters A, B, C, D, and E.The number of formal interviews was 10, 9, 9, 10 and 10 respectively.

## 2.3.  Data collection

This study used a semi-structured interview to interview family caregivers face-to-face.The patient's medical record was reviewed before the interview to understand the patient's current medical history, past history, examination results, treatment and care. With the assistance of a expert who had done qualitative research, the interviewer explained the purpose, significance, and privacy protection of the interview to the respondent, and obtained informed consent. And we were committed to protecting privacy and using data only for scientific research.

Each interviewees had a scheduled appointment in advance to ensure that the interviews were conducted without interruptions or distractions. Before the interview began, each interviewees provided informed consent, and demographic data were collected. The interview guide was developed based on the research objectives, the theoretical framework of Timing It Right theory, and a thorough review of relevant literature. The guide was designed to address key themes related to the caregivers' experiences and needs, while simultaneously allowing for the exploration of emergent topics not initially anticipated by the researchers. Prior to the formal interviews, three pilot interviews were conducted to test the feasibility and reliability of the interview guide, ensuring that the research process was well-prepared and refined before full implementation.

The research team consisted of nursing managers, health managers, and clinical nurses, one of whom had experience in caring for family members. The researcher was responsible for the interview communication with the study subjects; one stroke health manager was responsible for recruiting the patients, recording the patients' expressions and movements during the interviews, and transcribing the interviews into text. The interview time was 30–60 min. Face-to-face interviews were mainly conducted in the study room and office of the department. Each interview was conducted in a quiet environment without the presence of irrelevant persons. All interviewees were interviewed only once, and no repeated interviews occurred. The interview guide o was shown in Table 1.

Table 1. The Interview Guide for Participants in the Five Various Phases.

| Phases | Questions |
|---|---|
| Diagnosis Phase | • How did you feel when you learned that your family member had a stroke?<br>• What difficulties have you encountered since the patient was admitted? |
| Stabilization Phase | • How has caring for a stroke family member in the hospital affected you?<br>• In the hospital to take care of patients during this time, what do you not adapt to? |
| Discharge Preparation Phase | • What help or guidance would you like your health care provider to provide you with to help the patient recover at home before the family member is discharged from the hospital? |
| Adjustment Phase | • What difficulties have you encountered while taking care of your family at home?How was it resolved?<br>• What kind of help do you expect the hospital or society to provide? |
| Adaptation Phase | • What are the remaining care difficulties currently?<br>• Any more needs? |

## 2.4. Ethics consideration

This study was approved by the Ethics Committee of Xiangtan Central Hospital (register NO.2023-09-013). All methods were performed in accordance with the consolidated criteria for reporting qualitative research (COREQ) [13]. The study complied with the Declaration of Helsinki; all participants signed informed consent forms and participated anonymously.

## 2.5. Rigor and reflexivity

The researchers were guided by Morse's strategies to ensure the rigor and credibility of the findings. For example, each interview followed a consistent process, using the same opening questions and using similar prompts to ask participants for more information. The interviews were conducted by the researcher (Miaozhen Wang) and supported by a neurology clinical nurse (Ke Wang). They all had received training in qualitative interview methods and knowledge of psychological care before the interviews. The principal investigator (Fangqun Cheng) was a nursing administrator trained in public healthhand, having the training and experience to conduct qualitative research. Ke Wang was an experienced qualitative researcher who trained others in qualitative methods and software. All researchers had no previous relationship with the participants.

The research team acknowledged potential biases related to their professional roles. Nursing administrators might prioritize systemic interventions, while clinical nurses could focus on practical caregiving challenges. To address this, regular reflective discussions and peer debriefing were conducted during data analysis. Emerging themes were cross-validated through member checking with participants, ensuring interpretations remained grounded in their perspectives.

During the interview, the interviewers remained neutral and avoided leading questions whilst observing and recording the participant's voice, mood changes, and body movements. Researchers only made appropriate follow-up questions and did not express their own views. Then tentative questions (e.g., 'Can you expound?') were added where necessary to fully understand the participants' needs. After each interview, the interviewer recorded the gains and losses of the interview in a reflective diary for improvement. Theresearch team was supervised throughout by a senior researcher (Fangqun Cheng).

## 2.6. Data analysis

To ensure the completeness and accuracy of the data, within 24 hours of each interview, the audio recordings were transcribed verbatim into text and organized. The data was imported into the NVivo 12 software subsequently. A triangulation approach was employed to enhance rigor: two researchers independently coded the data using Braun and Clarke's thematic analysis method [14], with discrepancies resolved through iterative discussions. If consensus was not reached, a third senior researcher with expertise in qualitative methods was consulted to review the data and finalize coding

decisions. The analytical process included six phases: (1) Immersive reading of all transcripts to gain a holistic under-standing of caregivers' experiences; (2) Extracting significant statements related to caregiving needs and challenges; (3) Open coding of recurring ideas to generate initial codes, followed by grouping codes into initial categories;(4) Assembling coded categories into preliminary themes; (5) Refining themes through constant comparison across transcripts to ensure coherence and distinctiveness; (6) Validating themes through peer debriefing with independent qualitative researchers and member checking. The initial coding was independently carried out by two authors, with a third author regularly moni-toring and intervening in case of differing opinions. Encoding attempts to identify and describe aspects and their outcomes were considered to constitute contextual or mechanistic features. Then, the fourth author reviewed all coding results. In the process of inductive coding and deductive classification, if there was any inconsistency, the research team engaged in discussions to achieve consensus, thereby ensuring the precision and accuracy of the study's outcomes. Finally, the research group discussed and reviewed all the identified themes and categories. We removed the participant identification and assigned each text an identification code to maintain the confidentiality of the participants.

## 3. Results

### 3.1. General information of participants

A total of 48 family caregivers of stroke with disabilities were selected to participate in the study. Among the caregivers, 12 were male and 36 were female, with a median age of 52. General information of all the participants is shown in Table 2.

### 3.2. Thematic analysis

The findings of this study were synthesized into a thematic framework (Fig 1) spanning the caregiving journey across five distinct phases: Diagnosis, Stabilization, Discharge Preparation, Adjustment, and Adaptation. This framework highlights how caregivers' roles evolve in response to both patient recovery trajectories and systemic constraints. Each phase is associated with specific themes that reflect the evolving needs and experiences of family caregivers of stroke patients with disabilities. Critically, themes transition from acute crisis management (diagnostic phase) to chronic adaptation (adaptation phase), with each phase's outcomes directly informing subsequent priorities.

#### 3.2.1. Diagnosis phase. **Theme 1 Hospital adaptation & examination navigation:** The development of Chinese medical caregivers remains at a nascent stage, Due to the influence of the traditional filial piety culture and the economic level of the family, most caregivers are required to take care of stroke patients throughout the entire phase of hospitalization from the onset of the stroke incapacitation. During this phase, the main caregiving tasks of caregivers are accompanying patients for examinations and assisting patients in their daily lives. However, caregivers were unfamiliar with the hospital environment and were uncertain about the number and intensity of family's examinations and treatments during the diagnosis phase. Many caregivers reported difficulties in adapting to the hospital environment and process. Therefore, there is an urgent need to adapt to the living condition of caring for patients in hospitals.

*"When we go for checkups, we don't know the place very well; in the middle of the journey, (carrying) a bed-ridden patient like him, it's a little bit difficult".(A5, young female caregiver with a slim build).*

*"It's also the first time I came to your hospital, where to do (checkups) and where to eat, I have to ask."(A9, middle-aged female, factory worker)*

**Theme2 Disease knowledge acquisition:** In face of the misfortune by a sudden stroke, most caregivers are at a loss and are unaware of disease-related content such as stroke treatment and prognosis, how to cope with the patient's existent condition and incapacitation. These realities force a strong need to learn about the disease and caregiving skills.

**Table 2. General information of all the participants.**

| Demographic characteristics | Total (n = 48) N(%) | Diagnosis (n = 10) N(%) | Stabilization (n = 9) N(%) | Preparation (n = 9) N(%) | Implementation (n = 10) N(%) | Adaptation (n = 10) N(%) |
|---|---|---|---|---|---|---|
| **Gender** | | | | | | |
| Male | 12 (25.0) | 0(0.0) | 3(33.3) | 2(22.2) | 5(50.0) | 2(20.0) |
| Female | 36 (75.0) | 10(100.0) | 6(66.7) | 7(77.8) | 5(50.0) | 8(80.0) |
| **Age stage** | | | | | | |
| 18–30 | 1 (2.1) | 1(10.0) | 0(0.0) | 0(0.0) | 0(0.0) | 0(0.0) |
| 31–40 | 6 (12.5) | 2(20.0) | 2(22.2) | 0(0.0) | 2(20.0) | 0(0.0) |
| 41–50 | 15 (31.3) | 2(20.0) | 3(33.3) | 5(55.6) | 4(40.0) | 1(10.0) |
| 51–60 | 15 (31.3) | 4(40.0) | 2(22.2) | 3(33.3) | 1(10.0) | 5(50.0) |
| 60+ | 11 (22.9) | 1(10.0) | 2(22.2) | 1(11.1) | 3(30.0) | 4(40.0) |
| **Education** | | | | | | |
| Less than high school | 29 (60.4) | 2(20.0) | 4(44.4) | 8(88.9) | 7(70.0) | 8(80.0) |
| High school | 12 (25.0) | 4(40.0) | 3(33.3) | 0(0.0) | 3(30.0) | 2(20.0) |
| University | 7 (14.6) | 4(40.0) | 2(22.2) | 1(11.1) | 0(0.0) | 0(0.0) |
| **Career** | | | | | | |
| Unemployment | 11 (22.9) | 3(30.0) | 2(22.2) | 2(22.2) | 4(40.0) | 0(0.0) |
| On-The-job | 28 (58.3) | 5(50.0) | 6(66.7) | 6(66.7) | 4(40.0) | 7(70.0) |
| Retirement | 9 (18.8) | 2(20.0) | 1(11.1) | 1(11.1) | 2(20.0) | 3(30.0) |
| **Place of residence** | | | | | | |
| Rural | 26 (54.2) | 5(50.0) | 2(22.2) | 6(66.7) | 8(80.0) | 5(50.0) |
| Urban | 22 (45.8) | 5(50.0) | 7(77.8) | 3(33.3) | 2(20.0) | 5(50.0) |
| **Caregiving experience** | | | | | | |
| Experienced | 14 (29.2) | 5(50.0) | 4(44.4) | 1(11.1) | 2(20.0) | 2(20.0) |
| No experience | 34 (70.8) | 5(50.0) | 5(55.6) | 8(88.9) | 8(80.0) | 8(80.0) |
| **Average daily care time** | | | | | | |
| 8h–12h | 16 (33.3) | 2(20.0) | 1(11.1) | 3(33.3) | 6(60.0) | 4(40.0) |
| 12h–16h | 1 (2.1) | 1(10.0) | 0(0.0) | 0(0.0) | 0(0.0) | 0(0.0) |
| 16h+ | 31 (64.6) | 7(70.0) | 8(88.9) | 6(66.7) | 4(40.0) | 6(60.0) |
| **Relationship with patients** | | | | | | |
| Spouse | 23 (47.9) | 4(40.0) | 4(44.4) | 2(22.2) | 5(50.0) | 8(80.0) |
| Son/daughter | 16 (33.3) | 4(40.0) | 2(22.2) | 5(55.6) | 3(30.0) | 2(20.0) |
| Other relatives | 9 (18.8) | 2(20.0) | 3(33.3) | 2(22.2) | 2(20.0) | 0(0.0) |
| **Stroke type** | | | | | | |
| Ischemic stroke | 33 (68.8) | 10(100.0) | 8(88.9) | 6(66.7) | 3(30.0) | 6(60.0) |
| Hemorrhagic stroke | 15 (31.2) | 0(0.0) | 1(11.1) | 3(33.3) | 7(70.0) | 4(40.0) |

*"A little bit anxious and then a little bit bewildered about the uncertainty of recovery." (A6, university-educated unemployed female).*

*"When he vomit, I don't know what to do, I can only call the doctor." (A3, elderly female patient's wife).*

*"It's when grandma eats, she chokes a lot and then I don't know how to help her." (A1, female high school student).*

**3.2.2. Stabilization phase. Theme1 Rehabilitation skill development:** In this phase, patient's diagnosis has been clarified, the condition has stabilized, and the rehabilitator will carry out the rehabilitation training according to the

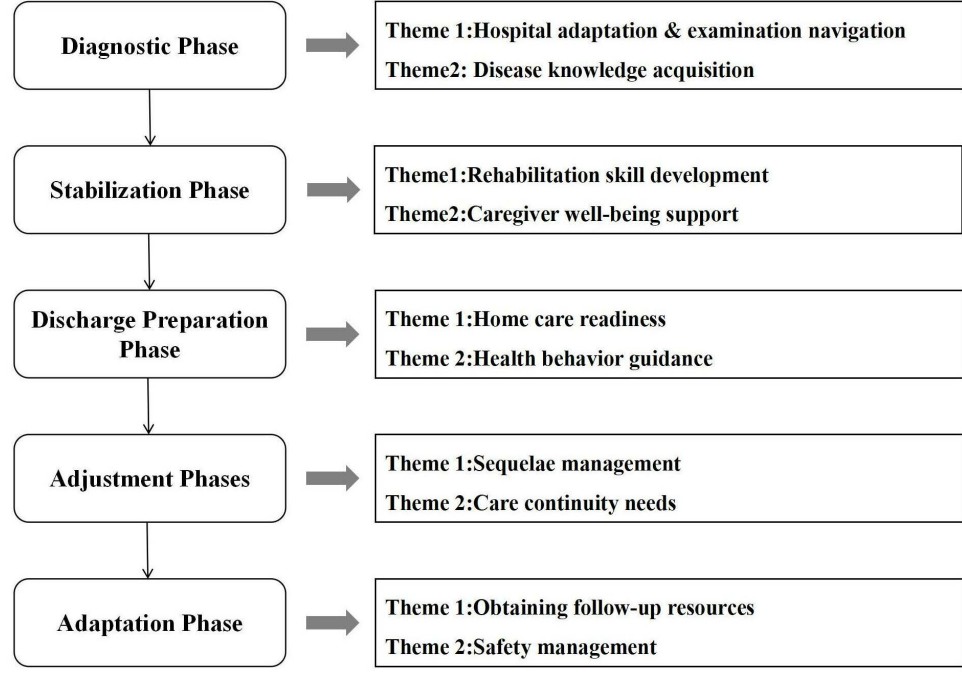

**Fig 1. Thematic framework of family caregivers' needs across different care phases for stroke patients with disabilities.**

patient's incapacity. Caregivers do not know the patient's rehabilitation knowledge and skills, and want to work with the rehabilitation team to ensure patient safety and promote recovery of the patient's functions.

*"I don't know much about rehabilitation either. When the rehabber comes over, I ask more questions and sometimes take a video to see how she does it." (B1, middle-aged patient's daughter with a university degree).*

*"Now he is in such a weak state, (getting out of bed and walking around) because I am afraid that he will fall down. As for the swallowing function, I don't know how to promote its recovery." (B5, 33-year-old patient's daughter-in-law).*

**Theme2 Caregiver well-being support:** Unlike nurses or paid companions who have extensive experience in caregiving, caregivers generally have difficulties in caregiving. Stroke disrupts caregivers' work and family lives, and some older caregivers report negative impacts on their own health. Caregivers expect to reduce the burden of caregiving and adjust to changes in their own life trajectories due to the family member's illness to minimize the negative impact of changes on the family as a whole.

*"When I was taking care of her, I really didn't know anything, and it took a few times of watching the nurses messing with the urinal to learn a little bit [to use the urinal to catch urine.It was really exhausting."(B2, soon-to-retire hypertensive female caregiver).*

*"Caring for so far, I'm slowly having trouble sleeping at night on my own...... Always thinking about him...... And back pain and bad eyes." (B3, older caregiver with a herniated disc).*

**3.2.3. Discharge preparation phase. Theme 1 Home care readiness**: In this phase, the patient is stabilized and is about to return home, with insufficient health care resources in the community. Caregivers are about to lose their quick access to caregiving knowledge in the hospital and are eager to learn about home care.

*"After being discharged from the hospital, I definitely want to know how to keep this disease from recurring? Now that he has undergone surgery, (pointing to his head with his finger) he has a stent in his brain, so how should I take care of him? In terms of diet, I also want to understand."*

*(C8, patient's uncle residing in a rural area)*

**Theme 2 Health behavior guidance**: Caregivers are eager to learn about stroke-related health-promoting behaviors and poor lifestyle behaviors to help patients improve health outcomes.

*"What activities are okay to do (after discharge)? Which ones can't be done? Which ones will promote his recovery? I wish someone would tell me all of them." (C7, university-educated tour guide).*

*"Can I still smoke and drink? My dad's favorite thing is to smoke and drink." (C6, junior high school-educated patient's daughter).*

**3.2.4. Adjustment phase. Theme 1 Sequelae management:** At this phase, China's community-based rehabilitation resources are inadequate, rehabilitation costs are high, and some families are unable to sustain long-term rehabilitation expenditures and need to rehabilitate at home. Patients still have some functional disabilities after discharge from the hospital, and caregivers need to know about rehabilitation in order to guide the patient's functional rehabilitation at home and improve the patient's sequelae.

*"His right hand is still a bit numb, and he is still tired of climbing the stairs, and I don't know how to do rehabilitation for him." (D5, 50-year-old caregiver with junior high education).*

*"The fingers of her right hand are unstable in holding chopsticks, and I don't know how to get them to work. I bought her an exerciser, but she'd rather go downstairs and play mahjong with other people." (D8, middle-aged male caregiver balancing work commitments).*

**Theme 2 Care continuity needs:** Some patients still have problems that were not fully resolved during hospitalization. And they are unaware of convenient access to continuity of care. Caregivers have an insufficient stock of relevant knowledge and need guidance on continuity of care.

*"What about that phlebitis with hardened blood vessels in the hands?" (D7, patient's wife in a rural household).*

*'The medication that was prescribed at discharge, does she need to take it every day now? Or does she not need to take them anymore?" (D10, 36-year-old unemployed caregiver).*

**3.2.5 . Adaptation phase. Theme 1 Obtaining follow-up resources:** At this phase, Chinese hospitals generally follow up discharged patients within one month. In contrast, stroke patients have high recurrence, and follow-up after three months of discharge is less frequent and simpler. Follow-up visits for stroke patients in the community, especially in rural areas are lacking. Caregivers, faced with patients' existing problems after discharge, do not know the ways or methods of follow-up and would like to have easy access to follow-up resources.

*"It would be nice if your hospital could visit us more often, to ask how things are going, to see how [the patient] is recovering from the symptoms, and to help us with our questions." (E1, 54-year-old caregiver with junior high education).*

*"I don't even know how to make a follow-up visit, is it the same as the usual outpatient visit?" (E7, 67-year-old caregiver with primary school education).*

*"If he has an emergency, then can I come directly to the hospital (inpatient unit) to see you then?"(E8, employed caregiver with primary school education).*

**Theme 2 Safety management:** In this phase, some patients' degree of incapacity improved and range of activities expanded after rehabilitation, and their family members' management of falls and wandering was more lax than before. However, the safety risks of patients' falls or wandering still existed, caregivers were eager to access to information about safety management.

*"When he (a person) goes out to walk, I'm so afraid that he'll fall." (E7, 67-year-old caregiver with primary school education).*

*"Do you guys have any recommendations for one of those professional locator things that she can find her if she goes out and gets lost. I bought her a bracelet online to wear and she just chewed it off and threw it away." (E9, retired, patient's daughter co-residing in a rural villa).*

## 4. Discussion

In this study, we explore the dynamic, whole-course care needs of family caregivers of stroke patients with disabilities in the Chinese cultural context based on the TIR theory framework, which reflects the temporal and contextual nature of caregivers' needs. The dynamic needs identified in this study emerge from the interplay between China's unique socio-cultural context and systemic healthcare challenges. In the diagnosis phase, caregivers primarily focus on adapting to the hospitalized environment and accompanying patients for examinations, while also seeking to acquire disease-related knowledge. During the stabilization phase, the emphasis shifts to enhancing caregivers' knowledge and skills for effective rehabilitation and addressing the negative impact on caregivers' well-being. In the discharge preparation phase, caregivers begin acquiring knowledge about home care and are eager to health behavior guidance. As the patient transitions home, the adjustment phase reveals that caregivers require information of managing sequelae and continuity of care. Finally, in the adaptation phase, caregivers seek follow-up resources and face challenges in safety management. We have found that caregivers' needs are more complex in the specific cultural context of China, particularly in terms of the desire for knowledge about the disease and psychological adjustment to caregiving responsibilities. This phenomenon has been less frequently emphasized in studies from other countries, suggesting that the needs of Chinese caregivers are distinctly geographically and culturally specific. These interpretations should be contextualized by methodological limitations. The single-site design may underrepresent rural caregiving realities, while cross-sectional data limit tracking of need evolution. Future multi-center longitudinal studies should be conducted to address these constraints.

### 4.1. Diagnosis phase: Introducing the hospital environment and the matters of examination, helping caregivers overcome their lack of knowledge dilemma

Confronted with the sudden onset of stroke in patients, family caregivers are often unprepared yet must rapidly adapt to the hospital's physical environment, comprehend complex examination protocols, and acquire substantial stroke-related knowledge within limited timeframes. Unlike Western contexts where hospital staff often manage patient logistics [9], Chinese caregivers assume dual roles as emotional supporters and logistical coordinators due to underdeveloped medical caregiver systems. This systemic gap, less emphasized in Western studies with established professional support frameworks, elucidates caregivers' prioritization of hospital adaptation, a need scarcely documented in high-resource settings that further reveals culturally embedded caregiving disparities [4].

Hospitals can deploy additional support staff, such as health care workers and volunteers, to assist caregivers in adapting to the ward and hospital environment and provide help with accompanying patients for examinations. Augmented reality-based indoor navigation systems can also be developed to help provide instructions on the correct location of the patient while caring for them in the hospital [15]. So hospitals should be mandated to assign transition navigators for families with elderly caregivers, to reduce logistical stress during the diagnosis phase.

This study also shows that caregivers in this phase exhibit a lack of theoretical knowledge, which is consistent with prior research [4]. It may be attributed to systemic gaps in caregiver education and the prioritization of immediate practical tasks over long-term knowledge acquisition in high-stress caregiving environments. Appropriate caregiving knowledge education [16] can be given according to the patient's symptoms and signs, including knowledge of the disease, medication, diet, exercise, psychological guidance, precautions and rehabilitation techniques. Additionally, study have suggested that offering caregivers video tutorials can significantly and safely improve their self-efficacy and awareness of the causes of stroke, enhance their caregiving skills [17]. Nurses can use all kinds of diversified means to meet the needs of their lack of knowledge, help them adapt to caring for patients hospitalized in the state of caregiving, such as creating instructional videos, using public science platforms, and distributing paper education sheets to caregivers.

### 4.2. Stabilization phase: Training in caregiving and rehabilitation skills, combining with support for caregivers

During this phase, caregivers are often uncertain about how to assist in the rehabilitation process, especially when the patient's condition has stabilized but rehabilitation continues, which is consistent with prior studies [8,18]. This uncertainty, rooted in fragmented support systems, drives their desire to collaborate with multidisciplinary rehabilitation teams to ensure the safety of the patient and promote functional recovery. Relevant studies have shown that structured training for caregivers can reduce their caregiving burden, improve quality of life, and reduce the length of hospitalization of patients [18]. According to previous research [19], nurse-mediated exercise programme has the potential to improve outcomes in terms of body function, activities, and participation in people with stroke. In addition to focusing on patient treatment, medical staff should do a good job of multidisciplinary teamwork to provide structured training for different caregivers on caregiving knowledge and rehabilitation cooperation, such as the use of printed manuals, hands-on demonstrations, instructional videos, or training sessions with rehabilitation professionals, etc.

This study reveals that caregivers' burden remained heavy and significant negative impacts on their personal well-being, particularly in relation to physical health, due to the increasing rehabilitation and caregiving demands. Many caregivers experienced fatigue, sleep disturbances, and physical strain, such as back pain and eye issues. While prior study [18] emphasize skill training, our findings uniquely highlight caregivers' physical strain during this phase. These findings align with previous research by Zarit [20]. To alleviate the physical and emotional toll on caregivers, it is essential that interventions be designed to both enhance caregivers' skills in caregiving and reduce the negative effects of caregiving on their own health. For instance, offering caregivers respite care services or support groups to address physical strain [21]. Additionally, providing regular respite opportunities or physical health services would help caregivers recover and manage the physical burdens of caregiving. Moreover, caregiver skills training programs should incorporate structured ergonomic modules that educate energy-saving techniques through biomechanical principles, including maintaining a widened stance with legs positioned shoulder-width apart to enhance stability during patient transfers, lowering the center of gravity through controlled hip and knee flexion instead of spinal bending, and utilizing assistive devices such as slide sheets or transfer belts to reduce musculoskeletal strain during repositioning tasks. Furthermore, municipal health bureaus could establish hospital-based respite centers through dedicated caregiver support grants, offering supervised care services to alleviate caregiver burden. Certified volunteer programs could integrate retired professionals into basic patient assistance roles under clinical oversight, as demonstrated by Japan's Silver Human Resource Centers model where retirees contribute to healthcare support while enhancing their own well-being [22].

### 4.3. Discharge preparation phase: Customized home care plans for caregivers and guidance on health behaviors

The preparation phase for hospital discharge is a critical time for patients to return to their families. Due to the limitations of medical resources and economic level, about 80% of stroke patients choose home rehabilitation after their condition is stabilized [3]. However, at this phase, stroke transitional care is often inefficient, leading to unmet needs of family caregivers of stroke in China, post-discharge complications and avoidable readmission rates of patients [1]. Our findings align with global evidence [23] that caregivers prioritize home care readiness and health behavior guidance during this phase. Yet, what distinguishes the Chinese context is the intersection of Confucian filial piety norms and systemic gaps in community rehabilitation infrastructure, which intensify caregivers' reliance on hospital-based support.

Multiple studies [24, 25] have shown that nurse-led models of transitional care can reduce readmission rates. During this phase, the implementation of a nurse-led discharge preparation plan and the development of a home care program can develop the caregiver's ability to navigate the health and social care system and activate self-management skills [23]. Discharge care planns should be based not only on an assessment of the patient's or care recipient's needs, but also on the needs and readiness gaps of family caregivers. Relevant departments should require medical institutions to make discharge care plans for families of stroke patients with disabilities before they are discharged from the hospital, in order to address the problems of insufficient discharge preparation and uneven distribution of community resources.

Nurses can provide a personalized discharge plan for improving discharge readiness. Specifically, they can provide a reliable guarantee for patients' smooth transition from the hospital to the home, including working with the rehabilitator to develop a home rehabilitation plan, enhancing caregivers' education on relevant health knowledge, and identifying available social support. Meanwhile, for patients with high risk of falling or falling out of bed, caregivers should be instructed to eliminate potential safety hazards and avoid risky events caused by environmental factors. In addition, study [26] has confirmed that behavioral instruction involving healthy lifestyles and medication management prior to discharge reduce readmission rates for stroke patients. Nurses can provide guidance on healthy behaviors prior to discharge using health education sessions, stroke information brochures, and personalized medication information sheets to help manage patients' hypertension, lower blood glucose, control lipids, stay safe in physical activity, make lifestyle changes, and prevent stroke-related complications.

### 4.4. Adjustment phase: Providing continuity of care and rehabilitation guidance

As stroke patients are left with varying degrees of sequelae such as hemiparesis and dysphagias [1,2], they may still have varying degrees of functional impairment after discharge from the hospital. Studies have shown that approximately 25%−50% of chronic stroke patients require caregiver assistance in performing daily activitiess [2]. During this phase, stroke family caregivers need arise around the sequelae of stroke patients and need to provide continuity of care and rehabilitation guidance to facilitate functional recovery [27]. Our findings highlight caregivers' truggles with inadequate rehabilitation knowledge and limited social support, exacerbating burnout and suboptimal recovery outcomes.

Unlike contexts with well-established community rehabilitation resources, Chinese urban-rural healthcare disparity exacerbates challenges for rural caregivers, who often lack access to specialized rehabilitation services. The lack of professional guidance and a sound rehabilitation plan often leads to poor caregiver knowledge, beliefs, and behaviors, resulting in poor home rehabilitation for stroke survivors [28].

Existing research predominantly emphasizes the implementation of structured rehabilitation programs to optimize post-stroke recovery outcomes. However, our findings reveal a distinct phenomenon in the Chinese context. Caregivers frequently rely on informal methods to address rehabilitation gaps, such as traditional massage and experience sharing. Nurses can help caregivers identify potential support sources that are valuable in the caregiving role, including other family members and friends, caregiver support groups, and community organizations. Nurses also can provide regular home follow-up visits, and assist caregivers with the patient's home rehabilitation and care.

However, for caregivers with inadequate social support, nurses can use telemedicine technology to assist in the patient's home rehabilitation and continuation of services, given the uneven distribution of existing community healthcare resources. A recent randomized controlled trial [29] demonstrated comparable efficacy between home-based telerehabilitation and conventional rehabilitation for stroke survivors, achieving equivalent improvements in activities of daily living (ADLs) and balance function, while attaining similar reductions in caregiver burden. This evidence underscores telerehabilitation's viability as a cost-effective alternative in resource-constrained healthcare systems. Multiple systematic evaluations have shown that telemedicine meets caregivers' home care needs, reduces rehabilitation costs, breaks down geographic constraints, and it is effective in enhancing patients' motor function, general quality of life [11,30,31]. Caregivers can use the technology to retrieve relevant health vocabulary to obtain appropriate health knowledge, they can also conveniently contact medical staff to solve some of the problems encountered in home care [11]. Furthermore, hospitals also can collaborate with primary healthcare institutions and village health workers to implement a hybrid rehabilitation model for stroke families in rural areas, integrating app-guided protocols with community-based demonstrations.

### 4.5. Adaptation phase: Providing long-term follow-up services to assist in safety management

The results of this study confirm that Chinese caregivers at this phase lack resources for long-term stroke follow-up and proper safety management. Caregivers still need information about ensuring patient safety, including fall prevention, medications, proper prescriptions, adequate nutritionmanagement of healthy. Both the European Stroke Action Plan and the Swedish National Stroke Care Guidelines recommend follow-up 3–6 months after stroke. However hospital follow-up systems in China prioritize acute care over chronic management due to funding constraints. Urban hospitals often lack incentives to extend services beyond 1 month, while rural clinics lack capacity altogether. These systemic limitations directly explain our findings: providing long-term follow-up services to assist in safety management is both a critical need and a feasible intervention to address gaps in post-acute care continuity.

Hospitals and community nurses at all levels should provide long-term follow-up resources for stroke families, provide caregivers with ongoing information on home care, and help caregivers manage the patient's safety properly. On the one hand, the follow-up time for stroke patients should be extended, the number of follow-up visits should be increased, phone calls and home visits should be made, and regular hospital follow-up visits should be conducted when necessary. During the follow-up visits, caregivers should be taught about prevention of recurrence and informed about the symptoms and treatment in case of recurrence. In conjunction with the patient's existing symptoms and medication use, the patient also should be informed of the time and method of the next outpatient follow-up visit. In addition, community and healthcare organizations can conduct safety assessments based on the patient's existing dysfunctions, help caregivers better identify the patient's remaining safety risks, urge them to take preventive measures, and evaluate their implementation [32]. Policymakers should incentivize hospitals to standardize follow-up protocols, ensuring continuity of care for stroke survivors and other chronic disease patients for 6–12 months post-discharge. Concurrently, primary healthcare workers should be trained to support home care management and conduct safety audits.While prior research emphasized caregiver education, our findings stress the need for systemic reforms. For example, safety management requires not just caregiver training but also environmental modifications (e.g., subsidized home safety audits). This aligns with the WHO's call for systemic integration of rehabilitation into health systems through leadership, governance, and multidisciplinary collaboration, as outlined in the 'Rehabilitation 2030' framework [33]. However, most interventions remain siloed, underscoring the gap between policy advocacy and fragmented implementation in low-resource settings.

## 5. Limitation

This study was conducted in the same healthcare organization, which may limit the generalizability and representativeness of the findings, particularly for rural populations or regions with distinct healthcare infrastructures. To address this, future multi-site studies should be conducted across diverse healthcare settings to capture contextual variations in

caregiver needs. Second, the reliance on purposive sampling might introduce selection bias, as caregivers with higher caregiving burdens may have been more likely to participate. Future studies could mitigate this risk by combining stratified purposive sampling with quantitative validation to ensure broader representativeness. Thrid, there may be limitations in the applicability of TIR theory in different cultural contexts. In the Chinese cultural context, the needs of family caregivers may be profoundly influenced by factors such as Confucian ethics and family responsibilities, which may not be fully adapted to the explanatory framework of TIR theory. Future studies should explore cultural adaptations of the TIR theory, such as incorporating family-level decision-making phases in Confucian societies. Then, only one interview is conducted with each respondent in this study. Data collection at a single point in time may not provide a full picture of caregiver needs. Longitudinal mixed-methods designs, combining repeated interviews with quantitative measure, could better track evolving needs influenced by disease progression or policy changes. Finally, this study may have focused primarily on the internal needs of family caregivers. The impact of the external environment, such as social support, availability of medical resources, and policy environment, on caregiver needs is not sufficiently considered. Expanding the scope through policy analysis and resource mapping could clarify macro-level impacts on caregiving needs.

## 6. Conclusion

This study comprehensively explores the dynamic and evolving needs of family caregivers for stroke patients with disabilities based on the TIR theory. Key findings emphasize the importance of culturally sensitive, phase-specific interventions. During the diagnosis phase, the measures involve introducing the hospital environment and the matters of examination, helping caregivers overcome their lack of knowledge dilemma. In the stabilization phase, the measures include training in caregiving and rehabilitation skills, combining with support for caregivers. For the discharge preparation phase, the measures cover customized home care plans for caregivers and guidance on health behaviors. During the adjustment phase, the actions focus on providing continuity of care and rehabilitation guidance. In the adaptation phase, the efforts emphasize providing long-term follow-up services to assist in safety management. By mapping caregivers' evolving needs across five critical care phases through culturally attuned support, these findings illuminate pathways to align family-centered care models with strained healthcare infrastructures. The proposed strategies hold practical value across diverse healthcare settings, particularly in contexts strained by overwhelmed medical systems and aging populations, offering scalable solutions to mitigate caregiver burden and systemic inefficiencies. Future research should expand on longitudinal approaches to capture the evolving nature of caregiver needs and explore innovative strategies for integrating support systems across diverse cultural contexts.

## Author contributions

**Conceptualization:** Miaozhen Wang, Ke Wang, Bin Xie, Xing Cao, Chenting Liu, Hongyu Fu, You Zhou, Yan Zhu, Fangqun Cheng.

**Data curation:** Miaozhen Wang, Ke Wang, Bin Xie, Lingling Song, Xing Cao, Yunhui Tong, Chenting Liu, Hongyu Fu, You Zhou, Qinqin Chen, Yan Zhu, Ling Zhu, Fangqun Cheng.

**Formal analysis:** Miaozhen Wang, Bin Xie.

**Investigation:** Ke Wang, Bin Xie, Lingling Song, Xing Cao, Yunhui Tong, Hongyu Fu, Ling Zhu.

**Methodology:** Miaozhen Wang, Ke Wang, Bin Xie, Dan Liu, Lingling Song, Xing Cao, Yunhui Tong, Chenting Liu, You Zhou, Qinqin Chen, Fangqun Cheng.

**Resources:** Dan Liu.

**Software:** Miaozhen Wang.

**Writing – original draft:** Miaozhen Wang.

**Writing – review & editing:** Miaozhen Wang, Ke Wang, Fangqun Cheng.

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
