## [Decision Letter · Decision Letter 0]

Dear Dr. Cheng,

Thank you for submitting your manuscript to PLOS ONE. After careful consideration, we feel that it has merit but does not fully meet PLOS ONE’s publication criteria as it currently stands. Therefore, we invite you to submit a revised version of the manuscript that addresses the points raised during the review process.

We look forward to receiving your revised manuscript.

Kind regards,

I Gede Juanamasta

Academic Editor

PLOS ONE

 [This work was funded by the Hunan Provincial Health Commission Project (202203073335 ), Department of Science and Technology of Hunan Province (2024ZK4233), Hunan Provincial People's Hospital Medical Union Special Project (2023YLT002), Natural Science Foundation of Hunan Province of China (2024JJ9561), Chinese Nursing Association Project (ZHKY202406), and National Key Clinical Specialties Major Specialty Program of the Healthcare Commission of Hunan Province (Z2023138),Xiangtan Medical Association Project (2023-xtyx-40).]. 

Reviewers' comments:

Reviewer's Responses to Questions

**Comments to the Author**

1. Is the manuscript technically sound, and do the data support the conclusions?

Reviewer #1: Yes

Reviewer #2: Partly

2. Has the statistical analysis been performed appropriately and rigorously?

Reviewer #1: Yes

Reviewer #2: N/A

3. Have the authors made all data underlying the findings in their manuscript fully available?

Reviewer #1: No

Reviewer #2: No

4. Is the manuscript presented in an intelligible fashion and written in standard English?

Reviewer #1: Yes

Reviewer #2: No

Reviewer #1: I have read the manuscript, and it presents an interesting and valuable study. The writing is clear and provides meaningful insights into the target group.

While the title effectively reflects the study's scope, it lacks scientific precision. I suggest rephrasing it as 'Understanding the Needs of Family Caregivers of Stroke Patients with Disabilities: A Phenomenological Study Using the Timing It Right Theory' to enhance clarity and conciseness."

The abstract:

It is well written, providing a clear and structured summary of the study.

Concerns:

• The Timing It Right (TIR) theory is introduced but could benefit from a very brief mention of its core premise in a phrase to guide unfamiliar readers.

• Check the grammar eg: "Caregivers’ experiences and needs were explored across five stages of care: diagnosis, stabilization, discharge preparation, adjustment, and adaptation." – suggested revision: "This study explored caregivers' experiences and needs across five key stages of care: diagnosis, stabilization, discharge preparation, adjustment, and adaptation."

• The conclusion could be more assertive in emphasizing the study's contributions.

Introduction:

• The problem statement currently focuses on the burden of caregiving but could be more explicit in defining the core issue (the need) that necessitates this research.

• The introduction mentions previous studies on caregivers' psychological burden, coping strategies, and social support. However, the specific gaps in existing literature should be more explicitly stated.

Methodology and tool:

• The study mentions member checking (participants reviewing transcripts), but triangulation (e.g., multiple researchers analyzing the data) should be clarified.

• It states that researchers had no prior relationships with participants, which is good. However, a reflexivity statement (acknowledging potential biases) would enhance transparency.

• The description of the study setting and participants' demographics helps, but a clearer discussion on how findings may apply beyond this sample would be beneficial.

The results:

• The five caregiving stages (diagnosis, stabilization, discharge preparation, adjustment, adaptation) are well structured. Each stage is linked to a major theme, reflecting the evolving needs of caregivers.

• Each theme is supported by subthemes, making it easier to follow how findings emerged.

• The subthemes highlight specific concerns, such as knowledge acquisition, rehabilitation skills, safety management, and emotional burden. Well done!

Discussion:

• Deepen interpretation of findings by explaining why they emerged.

• Engage more critically with previous studies—highlighting what is new or different.

• Provide clearer, stage-specific recommendations for practice and policy.

• Expand the limitations section to address biases and generalizability.

• Strengthen the conclusion by emphasizing the study’s unique contributions.

Reviewer #2: Abstract

Specify the cultural context in the objective statement.

Briefly mention cultural influences in the results.

Add a sentence on practical implications.

Introduction

Streamline statistical data.

Introduce TIR theory earlier.

Improve transition between stroke burden and caregiver focus.

Methods

Clarify participant recruitment and sample size rationale.

Provide details on theme identification and validation.

Discuss reflexivity and researcher influence.

Results

Present themes more concisely.

Add demographic context for quotes.

Reference and describe Fig. 1 in detail.

Discussion

Critically address study limitations (e.g., single-site bias, lack of longitudinal data).

Expand on policy and healthcare implications.

Strengthen telemedicine discussion with specific studies/examples.

Limitations

Discuss the impact on generalizability.

Suggest ways to address limitations in future research.

Conclusion

Specify practical implementation steps.

Highlight broader societal impact.

General Comments

Ensure consistency in terminology and formatting.

Consider adding a demographic summary table.

Deepen the discussion of study limitations.

**Do you want your identity to be public for this peer review?** For information about this choice, including consent withdrawal, please see our Privacy Policy

Reviewer #1: **Yes: ** Mohd Ismail

Reviewer #2: **Yes: ** Yupin Aungsuroch

---

## [Author Response · Author response to Decision Letter 1]

17 Apr 2025

Reviewer #1: I have read the manuscript, and it presents an interesting and valuable study. The writing is clear and provides meaningful insights into the target group.

Answer: Thank you for your positive assessment of our work.

We sincerely appreciate your positive feedback and recognition of the significance and clarity of our work. Your acknowledgment of the study's value and its contribution to understanding the target population is highly encouraging. We have carefully addressed all comments from the reviewers to further strengthen the manuscript. Thank you for your time and constructive evaluation.

While the title effectively reflects the study's scope, it lacks scientific precision. I suggest rephrasing it as 'Understanding the Needs of Family Caregivers of Stroke Patients with Disabilities: A Phenomenological Study Using the Timing It Right Theory' to enhance clarity and conciseness."

Answer: Thank you for your constructive suggestion.

We have revised the title as recommended to better align with the study’s theoretical framework and methodological approach. This modification enhances both clarity and scientific rigor, as noted in your feedback.

The abstract:It is well written, providing a clear and structured summary of the study.

Answer: Thank you for your kind acknowledgment of the abstract’s clarity and structure.

We aimed to succinctly convey the study’s purpose and key findings, and we are pleased this resonated with you.

Concerns:

• The Timing It Right (TIR) theory is introduced but could benefit from a very brief mention of its core premise in a phrase to guide unfamiliar readers.

Answer:Thank you for this valuable suggestion.

We have integrated a concise definition of the TIR theory into the revised abstract:"...based on the framework of the 'Timing It Right' (TIR) theory, a conceptual model emphasizing dynamic support aligned with caregivers' evolving needs across distinct care phases."This addition enhances clarity and contextualizes the theoretical foundation of the study.

• Check the grammar eg: "Caregivers’ experiences and needs were explored across five stages of care: diagnosis, stabilization, discharge preparation, adjustment, and adaptation." – suggested revision: "This study explored caregivers' experiences and needs across five key stages of care: diagnosis, stabilization, discharge preparation, adjustment, and adaptation."

Answer:We appreciate your meticulous attention to detail.

The sentence has been revised as suggested to improve grammatical structure and readability.All similar instances in the manuscript have been checked and revised to ensure consistency.

• The conclusion could be more assertive in emphasizing the study's contributions.

Answer: Thank you for this suggestion.

We have expanded the conclusion to emphasize practical contributions:"Practical implications include integrating multidisciplinary teams, leveraging telemedicine for continuity of care, and designing caregiver education programs aligned with Confucian family values."This addition clarifies how the findings can inform clinical practice and policy development.

Introduction:

• The problem statement currently focuses on the burden of caregiving but could be more explicit in defining the core issue (the need) that necessitates this research.

Answer: Thank you for your constructive suggestion.

We have revised the problem statement to explicitly define the core issue as the dynamic, phase-specific needs of caregivers. We have removed some contents related to the burden of care. And we have added some information about the needs of caregivers in the first and second paragraphs. These revisions have focused the core issue of the research on the needs of caregivers.

• The introduction mentions previous studies on caregivers' psychological burden, coping strategies, and social support. However, the specific gaps in existing literature should be more explicitly stated.

Answer: Thank you for your feedback.

We have explicitly highlighted gaps in prior research, including the static/fragmented perspectives and the lack of attention to cultural mediation (e.g., Confucian norms) and systemic challenges (e.g., urban-rural disparities). These additions are integrated into the fourth paragraph of the Introduction.

Methodology and tool:

• The study mentions member checking (participants reviewing transcripts), but triangulation (e.g., multiple researchers analyzing the data) should be clarified.

Answer: Thank you for your recommendation.

We clarified triangulation by specifying dual independent coding by researchers and third-party arbitration for discrepancies. This is detailed in the "Data Analysis" section under the revised Methods.

• It states that researchers had no prior relationships with participants, which is good. However, a reflexivity statement (acknowledging potential biases) would enhance transparency.

Answer: Thank you for your constructive suggestion.

 We have added a reflexivity statement in the "Rigor and Reflexivity" section to explicitly acknowledge potential biases arising from the research team’s professional roles (e.g., nursing administrators prioritizing systemic interventions versus clinical nurses focusing on practical challenges). This statement also outlines mitigation strategies, such as regular reflective discussions and peer debriefing, to ensure balanced interpretations. The revision enhances transparency by addressing how researcher perspectives were critically examined, aligning with your recommendation.

• The description of the study setting and participants' demographics helps, but a clearer discussion on how findings may apply beyond this sample would be beneficial.

Answer:Thank you for your constructive suggestion.

We have revised the "Participants and Sampling" section to explicitly discuss the potential applicability of findings beyond the study sample.These additions appear in the revised manuscript under the "Participants and Sampling" subsection, addressing how the study’s design and sample diversity support broader applicability. We stated that the diversity of sample enhances the potential applicability of findings to caregivers in similar socioeconomic and cultural contexts within China.

The results:

• The five caregiving stages (diagnosis, stabilization, discharge preparation, adjustment, adaptation) are well structured. Each stage is linked to a major theme, reflecting the evolving needs of caregivers.The subthemes highlight specific concerns, such as knowledge acquisition, rehabilitation skills, safety management, and emotional burden. Well done!

Answer:Thank you for your recognition of the thematic framework design.

The stages were carefully structured to reflect caregivers’ dynamic needs, and your positive feedback reinforces the validity of this approach.

Discussion:

• Deepen interpretation of findings by explaining why they emerged.

Answer: Thank you for your constructive suggestion.

In the discussion section, we elaborated on the findings of each stage and provided explanations to enhance the overall understanding and significance of the research. Additionally, according to your suggestion, we compared our findings with previous studies to identify similarities and discrepancies, which helped in validating our conclusions and suggesting potential areas for future exploration.

• Engage more critically with previous studies—highlighting what is new or different.

Answer: Thank you for your feedback.

We contrasted findings with prior studies, emphasizing novel themes like hospital navigation challenges and reliance on informal rehabilitation methods in China. For instance, the revised Discussion highlights how caregivers’ use of traditional massage diverges from structured programs in high-resource settings. These comparisons are integrated throughout the Thematic Analysis subsections.

• Provide clearer, stage-specific recommendations for practice and policy.

Answer: Thank you for your notice.

We refined recommendations to align with each care phase. Examples include:Diagnosis Phase: Augmented reality navigation systems for hospital logistics.Adaptation Phase: Policy reforms for subsidized home safety audits.These targeted strategies are detailed in the Conclusion and respective phase subsections.

• Expand the limitations section to address biases and generalizability.

Answer: Thank you for your notice.

We deepened the analysis of potential biases (e.g., purposive sampling favoring high-burden caregivers) and proposed combining stratified purposive sampling with quantitative validation to enhance representativeness. This addition strengthens the Limitations discussion.

• Strengthen the conclusion by emphasizing the study’s unique contributions.

Answer: Thank you for your recommendation.

We underscored the study’s strategies hold practical value across diverse healthcare settings. These contributions are explicitly stated in the Conclusion subsection.

Reviewer #2:

Abstract:

Specify the cultural context in the objective statement.

Answer:Thank you for your feedback.

The Objective statement has been revised to highlight the cultural context:"This study aims to explore the comprehensive care needs of family caregivers of stroke patients with disabilities in the Chinese cultural context, based on the framework..."This modification underscores the study’s focus on cultural influences unique to China.

Briefly mention cultural influences in the results.

Answer: Thank you for your suggestion.

We have explicitly highlighted cultural influences on caregivers' needs in the Results:

"Caregivers' needs were influenced by cultural expectations (e.g., filial piety norms requiring family-based caregiving), resource limitations, and their evolving role over time."This addition clarifies how Confucian cultural values directly shaped caregivers' responsibilities and demands, such as prioritizing family-led care over external support services.

Add a sentence on practical implications.

Answer: Thank you for this suggestion.

We have expanded the Conclusion section to emphasize practical implications:"Practical implications include integrating multidisciplinary teams, leveraging telemedicine for continuity of care, and designing caregiver education programs aligned with Confucian family values."This addition clarifies how the findings can inform clinical practice and policy development.

Introduction:

Streamline statistical data.

Answer: Thank you for your suggestion.

We removed redundant global cost data (e.g., $89.1 billion GDP impact) and emphasized rising stroke incidence trends ("The absolute number of incident strokes globally increased by 70.0% over three decades"). This revision appears in the first paragraph of the Introduction.

Introduce TIR theory earlier.

Answer: Thank you for your input.

We introduced the TIR theory at an appropriate position in the second paragraph in advance, improving logical coherence.

Improve transition between stroke burden and caregiver focus.

Answer: Thank you for your suggestion. 

A restructured logic chain now connects epidemiological data to caregiver challenges: "imposing significant caregiving burdens..." transitions from economic impacts to familial roles, while "disproportionately bear" highlights inequitable burden distribution (Introduction, para 1). The subsequent TIR theory section bridges disease progression and dynamic caregiver needs, establishing a theoretical scaffold for the study’s focus.

Methods:

Clarify participant recruitment and sample size rationale.

Answer: Thank you for your feedback.

We clarified recruitment by specifying thematic saturation as the stopping criterion and maximum variation sampling to ensure diversity. This is included in the revised "Participants and Sampling" section.

Provide details on theme identification and validation.

Answer: Thank you for your feedback.

We have elaborated the details of theme validation steps, including peer debriefing and member checking, in the "Data Analysis" section. The six-phase analytical process is explicitly outlined.

Discuss reflexivity and researcher influence.

Answer: Thank you for your input.

We discussed how professional roles (e.g., nursing administrators vs. clinical nurses) might influence interpretations and mitigated this through regular reflective discussions and peer debriefing. This is addressed in the "Rigor and Reflexivity" section.

Results:

Present themes more concisely.

Answer: Thank you for your constructive suggestion.

We have streamlined the thematic titles and descriptions to enhance clarity. Redundant explanations in theme narratives were removed, focusing on core caregiver challenges. These revisions mainly appeared in the diagnosis phase and stabilization phase under "Thematic Analysis" subsections.

Add demographic context for quotes.

Answer: Thank you for your feedback.

We have appended demographic details to all participant quotes, including age, gender, occupation, and caregiving role. For example:"When he vomit, I don’t know what to do..." (A3, elderly female patient's wife)These additions contextualize quotes within caregivers’ lived experiences and are integrated throughout the "Thematic Analysis" subsections.

Reference and describe Fig. 1 in detail.

Answer: Thank you for your observation.

We expanded the description of Fig. 1 to clarify its role in first paragraph of the "Thematic Analysis" subsections. Added: "This framework highlights how caregivers' roles evolve in response to patient recovery trajectories and systemic constraints, transitioning from acute crisis management to chronic adaptation."

Discussion:

Critically address study limitations (e.g., single-site bias, lack of longitudinal data).

Answer: Thank you for your constructive suggestion.

We have critically addressed methodological constraints in both the Discussion and Limitations sections. In the concluding sentences of the Discussion's first paragraph, we explicitly state: "These interpretations should be contextualized by methodological limitations. The single-site design may underrepresent rural caregiving realities, while cross-sectional data limit tracking of need evolution. Future multi-center longitudinal studies could address these constraints." Additionally, the Limitations subsection now elaborates on strategies to mitigate single-site bias and proposes mixed-methods longitudinal designs for future validation, fully aligning with your recommendations.

Expand on policy and healthcare implications.

Answer: Thank you for your input.

We added policy-level recommendations, such as incentivizing hospitals to standardize follow-up protocols and training rural healthcare workers. The proposed implications are systematically integrated into the corresponding phase-specific subsections of the Discussion, where each care phase explicitly addresses how findings translate into actionable strategies.

Strengthen telemedicine discussion with specific studies/examples.

Answer: Thank you for your recommendation.

We incorporated specific examples, including a randomized controlled trial on home-based telerehabilitation demonstrating reduced caregiver burden (P < 0.001). This addition, along with references to systematic reviews, appears in the Adjustment Phase discussion.

Limitations:

Discuss the impact on generalizability.

Answer: Thank you for your constructive suggestion.

We expanded the discussion on generalizability by explicitly noting that findings may not fully apply to rural populations or regions with distinct healthcare infrastructures. This revision appears in the opening paragraph of the Limitations section.

Suggest ways to address limitations in future research.

Answer: Thank you for your feedback.

We added actionable recommendations for future studies, such as conducting multi-site research and employing longitudinal mixed-methods designs , to address sampling bias and contextual variability. These suggestions are integrated into the revised Limitations section.

Conclusion

Specify practical implementation steps.

Answer: Thank you for your suggestion.

We specified phase-specific strategies under the revised Conclusion, aligning r

---

## [Decision Letter · Decision Letter 1]

Understanding the Needs of Family Caregivers of Stroke Patients with Disabilities: A Phenomenological Study Using the Timing It Right Theory

PONE-D-25-04561R1

Dear Dr. Cheng,

We’re pleased to inform you that your manuscript has been judged scientifically suitable for publication and will be formally accepted for publication once it meets all outstanding technical requirements.

Kind regards,

I Gede Juanamasta

Academic Editor

PLOS ONE

Additional Editor Comments (optional):

Reviewers' comments:

Reviewer's Responses to Questions

**Comments to the Author**

Reviewer #1: All comments have been addressed

Reviewer #2: All comments have been addressed

2. Is the manuscript technically sound, and do the data support the conclusions?

Reviewer #1: Yes

Reviewer #2: Yes

3. Has the statistical analysis been performed appropriately and rigorously?

Reviewer #1: Yes

Reviewer #2: Yes

4. Have the authors made all data underlying the findings in their manuscript fully available?

Reviewer #1: No

Reviewer #2: No

5. Is the manuscript presented in an intelligible fashion and written in standard English?

Reviewer #1: Yes

Reviewer #2: Yes

Reviewer #1: Your thoughtful revisions and detailed responses to the reviewer comments have significantly strengthened the manuscript. It is a timely and meaningful contribution to the literature on caregiver support, particularly within culturally specific contexts. Well done on your excellent work.

Reviewer #2: Authors have addressed all the comments. I have no more comments.

Article is well-discribed. It is ready to be published.

**Do you want your identity to be public for this peer review?** For information about this choice, including consent withdrawal, please see our Privacy Policy

Reviewer #1: **Yes: ** Mohd Ismail

Reviewer #2: **Yes: ** Yupin Aungsuroch

---

## [Editor Report · Acceptance letter]

PONE-D-25-04561R1

PLOS ONE

Dear Dr. Cheng,

I'm pleased to inform you that your manuscript has been deemed suitable for publication in PLOS ONE. Congratulations! Your manuscript is now being handed over to our production team.

Kind regards,

on behalf of

Dr. I Gede Juanamasta

Academic Editor

PLOS ONE